# Metaproteomics reveals enzymatic strategies deployed by anaerobic microbiomes to maintain lignocellulose deconstruction at high solids

Payal Chirania[1,2,3,6], Evert K. Holwerda [3,4,6], Richard J. Giannone [1,3], Xiaoyu Liang[4], Suresh Poudel[1], Joseph C. Ellis[1,3], Yannick J. Bomble[3,5], Robert L. Hettich [1,3 ✉] & Lee R. Lynd[3,4 ✉]

Economically viable production of cellulosic biofuels requires operation at high solids loadings —on the order of 15 wt%. To this end we characterize Nature's ability to deconstruct and utilize mid-season switchgrass at increasing solid loadings using an anaerobic methanogenic microbiome. This community exhibits undiminished fractional carbohydrate solubilization at loadings ranging from 30 g/L to 150 g/L. Metaproteomic interrogation reveals marked increases in the abundance of specific carbohydrate-active enzyme classes. Significant enrichment of auxiliary activity family 6 enzymes at higher solids suggests a role for Fenton chemistry. Stress-response proteins accompanying these reactions are similarly upregulated at higher solids, as are β-glucosidases, xylosidases, carbohydrate-debranching, and pectin-acting enzymes—all of which indicate that removal of deconstruction inhibitors is important for observed undiminished solubilization. Our work provides insights into the mechanisms by which natural microbiomes effectively deconstruct and utilize lignocellulose at high solids loadings, informing the future development of defined cultures for efficient bioconversion.

[1] Biosciences Division, Oak Ridge National Laboratory, Oak Ridge, TN, USA. [2] Graduate School of Genome Science and Technology, University of Tennessee, Knoxville, TN, USA. [3] Center for Bioenergy Innovation, Oak Ridge National Laboratory, Oak Ridge, TN, USA. [4] Dartmouth College – Thayer School of Engineering, Hanover, NH, USA. [5] National Renewable Energy Laboratory, Golden, CO, USA. [6] These authors contributed equally: Payal Chirania, Evert K. Holwerda. ✉ email: hettichrl@ornl.gov; lee.r.lynd@dartmouth.edu

Biological production of liquid fuels from lignocellulosic feedstocks is of high interest as society navigates a transition away from fossil resources[1]. However, the recalcitrant character of such feedstocks impedes biological conversion and represents a major cost barrier[2,3]. One-step consolidated bioprocessing (CBP) without added enzymes, mediated by defined cultures of anaerobic lignocellulose-fermenting bacteria, is a promising strategy for converting lignocellulose to liquid fuels at low cost[4]. Because substantial titers of liquid fuel products are required to avoid high costs for product recovery and fermentation, biological processes for conversion of lignocellulose need to operate at high solids loadings—typically on the order of 15 wt% or more[5–7]. Around two-thirds of the mass content of lignocellulose is carbohydrate. An efficient sugar-to-liquid-biofuel microbial metabolism can achieve an end-product at 50% yield. Not considering titer restrictions and solids handling issues, 150 g/L solids loading would result in a maximum biofuel titer for ethanol of ~50 g/L.

For both enzymatic hydrolysis mediated by fungal cellulase[8–10] and for anaerobic digestion mediated by undefined microbial consortia[11–13], fractional carbohydrate solubilization is relatively constant with increasing solids until little to no free water remains, above which rates decline[14,15]. The solids loading at which diminishing solubilization is observed varies from system to system but is generally in the range of 150–180 g/L (15 to 18 wt%)[16–19]. However, for defined culture systems—e.g., those that might be used for the production of compounds other than methane, $H_2$, and mixed organic acids from cellulosic biomass via CBP—carbohydrate solubilization has been observed to decline with increasing solids at much lower loadings, e.g., <80 g/L[20,21]. The basis for this decline is unknown, although controlled experiments indicate that it is not explained by inhibition by fermentation products or inadequate growth media[22,23]. Hence, the diagnosis and remediation of declining lignocellulose solubilization is important to advance CBP toward commercial application.

Deconstruction of lignocellulose typically involves dozens of proteins with diverse functions and structures[24–26]. Undefined microbial consortia such as those occurring in anaerobic digestion systems represent a wealth of diversity[27–29], both in terms of organisms as well as the carbohydrate active enzymes (CAZymes)[30,31] they express. The composition and functional characterization of lignocellulose-fermenting microbiomes has been explored over the last decade using emergent omic technologies, including metagenomics, metaproteomics, and metatranscriptomics[32,33]. These techniques have been used to document up to a million protein-encoding sequences[34–36], tens of thousands of genes and transcripts encoding CAZymes[36–38] from over 200 gene families[36], and over 1000 identified microbial species[39,40]. Detailed inventories of CAZymes have been characterized from diverse environments including the rumen[37], anaerobic digesters[41], termites[42,43], and the moose[38] among others. Although most studies have not examined the spatial location of microbes and enzymes, Kougias et al.[34] differentiated enzymes and organisms present in the planktonic and substrate-adhered phases. Liang et al.[44] inventoried changes in CAZyme and microbiome composition for thermophilic, anaerobic digestion of switchgrass at various residence times. Abbassi-Guendouz et al.[45] performed metagenomic and phylogenetic analysis of mesophilic anaerobic digestion of shredded cardboard with solids concentration varying from 10 to 30 wt%. Most studies to date have focused on the metabolic potential of these communities (metagenomics), with only a handful of expression-based studies (metatranscriptomics) that have been mostly confined to single batch conditions[41,46,47]. Metaproteomics is uniquely suited to measure the physical presence, abundance, and location of both intracellular and extracellular enzymatic machinery and how they differ across conditions. Thus, metaproteomics provides complementary information to metagenomic or metatranscriptomics, but has not been reported for anaerobic lignocellulose-fermenting microbiomes as a function of solids loading.

In this study, we employ LC-MS/MS-based metaproteomic measurements in order to gain insight into mechanisms of lignocellulose deconstruction at solids loadings representative of those anticipated in an industrial process. An anaerobic, thermophilic, semi-continuously fed, methanogenic microbial enrichment cultivated over an extended period (550 days), referred to here as a lignocellulose-fermenting microbiome, is sampled at various solids loadings at steady-state and fractionated to identify key microbes and/or enzymes. We document solubilization performance with increasing solids, changes in the abundance of CAZymes across fractions, and the details of the methanogenesis pathways. The data and results described in this paper are an extension of the previous work described in Liang et al.[44], where different residence times were examined (20 to 3.3 days) at one fixed solids loading of 30 g/L. The resulting metagenomes from that work are used as a basis for the new metaproteomics analysis described in this paper, where the residence time is fixed at 10 days, with increasing solids loading from 30 to 75, 120, and finally 150 g/L of the same feedstock.

## Results

**Microbiome maintains undiminished solubilization at higher solids.** An anaerobic lignocellulosic thermophilic (55 °C) methanogenic microbiome was established by enriching an anaerobic digester inoculum on 30 g/L unpretreated June-harvested non-senescent switchgrass and incubating the culture for over 120 days before entering the data collection phase. The microbiome with a 1 L working volume was operated semi-continuously at a 10-day residence time by daily withdrawal of 100 mL of well-mixed bioreactor contents followed by addition of 100 mL total volume of switchgrass and growth media[44]. The bioreactor was operated at solids loadings of 30, 75, 120, and 150 g/L, with the cultivation at each loading lasting for at least 50 days, leading to an overall microbiome cultivation time of 550 days (Fig. 1A, Supplementary Table 1, and Methods).

The fraction of carbohydrate solubilized at steady state—quantified by the loss of glucose, xylose, and arabinose moieties in the solids—remained relatively constant at ~66.8% regardless of solids loading over the range considered. The rate of carbohydrate solubilization increased in proportion to solids loading (Fig. 1B). The main products observed were methane ($CH_4$) and carbon dioxide ($CO_2$), whereas volatile fatty acids (VFA) and hydrogen ($H_2$) were not detected at steady state (Supplementary Fig. 1, Supplementary Table 2). Methane and $CO_2$ measured in effluent gas ranged from 48.7 to 49.9%, and 47.2 to 49.0% respectively. Nitrogen was also present at < 5%, likely introduced in the feeding and sampling process. Volumetric gas production and the methane production rate increased proportionally with increasing solids loading (Fig. 1C, Supplementary Table 2), indicating that product formation by the microbiome was not differently affected up to 150 g/L solids.

**Fractionation reveals spatial distribution of enzymatic categories.** A metaproteomic analysis was undertaken at each solids loading to provide a detailed molecular view of the composition and function of the lignocellulose-fermenting microbiome. The overall amount of microbial biomass, as assessed by an initial screen of unfractionated microbiome via 1D LC-MS/MS-based metaproteomic analysis, did not increase proportionally (p-value > 0.05, two-tailed Welch's t-test) with the 5-fold increase

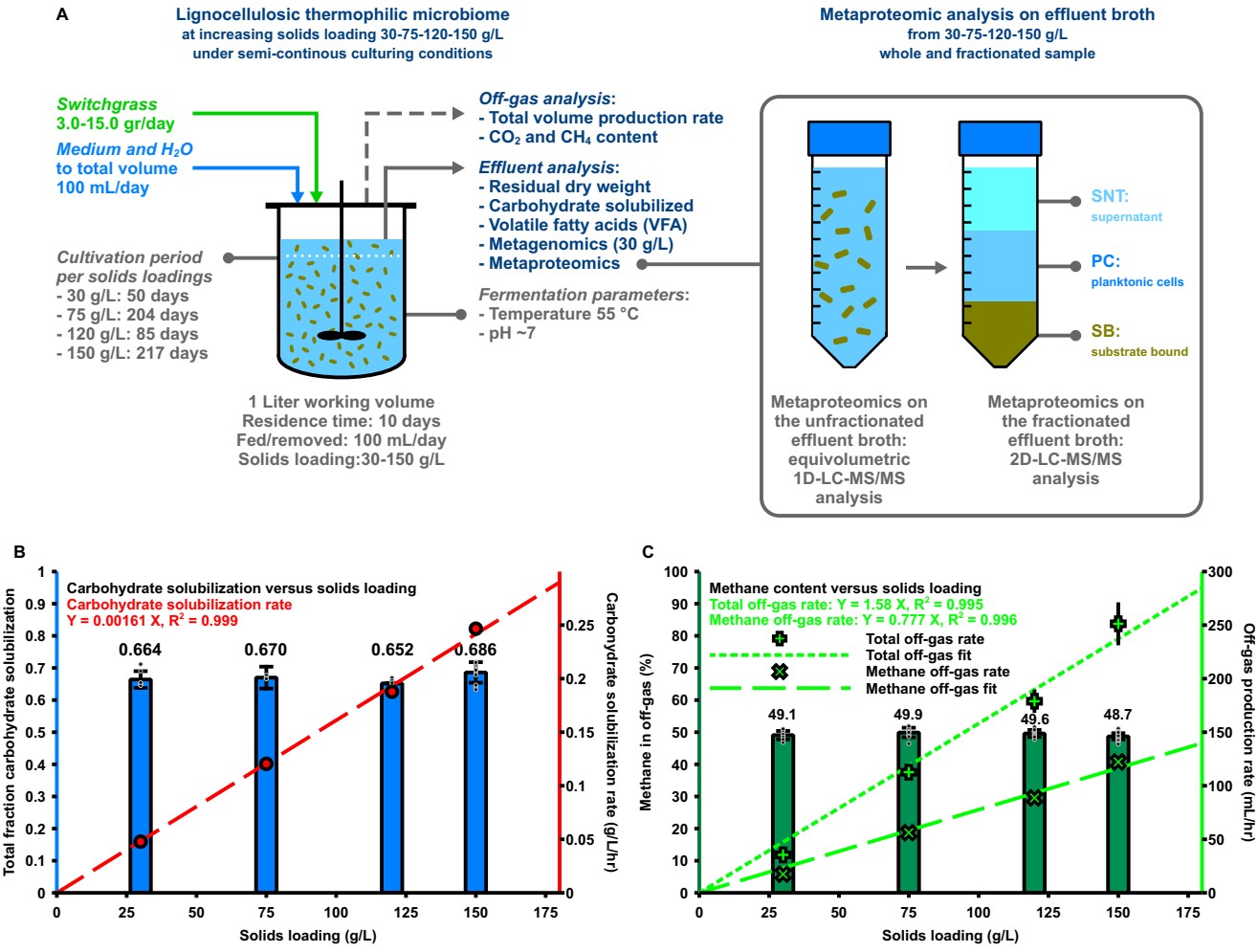

**Fig. 1 An anaerobic thermophilic microbiome exhibits undiminished fractional carbohydrate solubilization with increasing solids loadings between 30 and 150 g/L. A** Experimental overview. A lignocellulose-fermenting microbiome was fed semi-continuously with increasing amounts of switchgrass. Samples were analyzed for carbohydrate solubilization and metaproteomics. Metaproteomic analysis was carried out two ways; whole sample analysis (1D LC-MS/MS) and sample fractionation followed by multidimensional (2D LC-MS/MS) analysis. Prior to multidimensional measurements, each sample was separated by centrifugation into supernatant (SNT), planktonic cells (PC), and substrate bound (SB) fractions. **B** Stable carbohydrate solubilization (blue bars) at ~66.8% was obtained, with the rate of substrate solubilization (red circles) increasing linearly with solids loading. Data for 30 g/L was partially published prior in[44]. The data represents average values for different time-points during steady-state conditions for one microbiome (see Supplementary Table 2), error bars represent ± one standard deviation. The individual data points for total fractional carbohydrate solubilization (blue bars) are shown as dots (black circles) for each solids loading. The carbohydrate solubilization rate linear fit line (red dashed line) is based on carbohydrate solubilization rate data (red circles), which is calculated from fractional solubilization data (blue bars) adjusted for residence time in hours, the total carbohydrate content of switchgrass and the switchgrass loading concentration (30, 75, 120, and 150 g/L respectively). **C** The methane content of the off-gas (dark green bars) was constant, with the rate of off-gas (light green plus-shaped markers) and methane production (light green crosses) increasing in proportion to solids loading. Data for 30 g/L was partially published prior. Data points for methane content and off-gas production rate are averages for different time-points during steady-state conditions for one microbiome (see Supplementary Table 2), all error bars represent ± one standard deviation. The individual data points for methane concentration in the off-gas (dark green bars) are shown as dots (black circles) for each solids loading. Methane off-gas rate data is based on methane concentration in the off-gas and the off-gas production rate. Fit lines for the off-gas production rate (light green dashed line) and the methane off-gas production rate (light green dotted line) are linearly fitted to the data. Source data are provided as a Source Data file.

in substrate concentration; although significant abundance changes across a myriad of functional categories were observed with increasing solids (Fig. 1A, Supplementary Note 1, Supplementary Data 1). In order to further examine the changes in the community's functional structure across solids loadings and deepen the overall analysis, a multidimensional metaproteomic measurement was conducted on fractionated microbiome samples. As follows, each sample was split into three fractions: supernatant (SNT), planktonic cells (PC), and substrate-bound (SB). Sample fractionation to segregate extracellular proteins from planktonic- and substrate-bound microbes and enzymes has

been demonstrated previously, enabling the exploration of metabolic variability due to spatial localization of enzymes and microbes[34,48].

A total of 16,644 microbial protein groups were quantified based on unique peptide evidence across the three fractions at each substrate loading (Supplementary Data 2–4). There were both complexity and proteomic differences between the fractions, as only a third of the proteins were common among all three (Supplementary Fig. 2A). However, within each fraction, a comparable number of proteins were quantified across all solids loadings (Supplementary Fig. 2B), suggesting complexity

differences were driven by the type of fraction rather than the solids loading. When considering proteomic changes within each fraction, replicates of each solids loading clustered together, while broader differences were observed between low (30 g/L) and high loadings (120, 150 g/L) (Supplementary Fig. 2C).

Since the methane production rate increased proportionally with increasing solids, we first examined methanogenesis pathway-related proteomic signatures across the three fractions and assessed whether the observed increase was corroborated by metaproteomic analysis (Supplementary Fig. 3–6, Supplementary Data 5, Supplementary Note 2). The PC and SB fractions, which both include whole cells, had a greater representation of enzymes mapping to the different methanogenic pathways than the SNT fraction (Supplementary Fig. 3A). All methanogenesis pathway enzymes exhibited a clear increasing trend in the PC fraction that correlated with increasing solids loadings. This fraction also had the greatest representation of enzymes involved in the methanogenesis process (Supplementary Figs. 3A, B and 5). Closer inspection revealed that the PC fraction contained the highest total abundance of proteins from *Euryarchaeota* such that this phylum contributed substantially to the PC metaproteome relative to the other fractions (Supplementary Fig. 3C). Among the archaeal proteins, both the aggregate abundance and diversity (protein count) of those mapping the hydrogenotrophic pathway markedly exceeded that of the acetoclastic pathway (Supplementary Figs. 3A and 7), suggesting this to be the major route for methanogenesis.

**A diverse *Firmicutes*-dominated CAZyme repertoire is expressed**. CAZymes are directly responsible for catalyzing the conversion of carbohydrates harbored in lignocellulose into a soluble form. To fully characterize this central process and detail how CAZymes from different organisms and/or phyla synergize to efficiently deconstruct lignocellulose at increased solids, the fractionated metaproteomes were analyzed in terms of CAZyme type, abundance, and location. Prior to this analysis, a survey of the entire metagenome was conducted using the dbCAN2 meta server[49] to annotate important CAZyme families and their functional activities. Figure 2 shows the overview of CAZymes identified in each fraction and their general trends with solids loading (Supplementary Data 6). In total, 551 CAZyme protein groups were quantified (representing 1,246 proteins or ~14% of the total CAZymes annotated in the metagenome), with differences evident across the fractions (Fig. 2A). Notably, only 75 CAZymes (~14%) were present in all three fractions, whereas the PC and SB fractions harbored the greatest number of unique CAZymes (Fig. 2A). These differences demonstrate the compartmentalization of distinct CAZymes in the microbiome system and the importance of measuring spatially resolved fractions. About twenty percent (110 of 551) of the CAZymes harbored a carbohydrate binding domain (CBM) or a cellulosomal domain (cohesin or dockerin) (Supplementary Fig. 8, Supplementary Data 6). The proportion of these affinity-conferring CAZymes declined in the SNT fraction (from ~60% to ~20%) with increasing solids, while minimal changes were observed in the PC (~30%) and SB (~50%) fractions (Supplementary Fig. 8, 9), suggesting that free enzymes are somewhat excluded from binding substrate at lower solids loadings and remain in the SNT until surfaces become available.

Taxonomically, members of the phyla *Firmicutes* (*Clostridia* followed by *Bacillus*), known for their cellulolytic capabilities in thermophilic environments, were the dominant contributors of these measured CAZymes. However, fraction-specific differences in the taxonomic lineages of the CAZymes were observed (Fig. 2B). For example, CAZymes from the phylum *Chloroflexi*

were more prevalent in the PC fraction while enzymes from the phylum *Thermotogae* were more represented in the SNT and SB fractions (Fig. 2B), suggesting the presence of functional niches. From a functional perspective, Glycoside Hydrolases (GHs) formed the most numerous and abundant CAZyme class irrespective of the fractions (Fig. 2B), in line with previous studies[50]. Other CAZyme classes—polysaccharide lyases (PLs), carbohydrate esterases (CEs), and glycosyl transferases (GTs)—appeared to be dominant in the substrate- or cell-associated fractions while auxiliary activities (AAs) were more represented in the extracellular fraction (Fig. 2B), further indicating the enrichment of enzymatic activities based on the cellular location.

Quantitative assessment of CAZyme classes revealed trends in their abundances from low to high solids loadings in all three fractions (Fig. 2C, Supplementary Fig. 9C). AAs formed a small component of the CAZyme repertoire; however, they exhibited a significant increase in abundance in all fractions with increasing solids loading. Meanwhile, the abundances of PLs and CEs both decreased significantly in the SNT fraction at higher solids (120 and 150 g/L) but exhibited opposing trends in PC and SB fractions, especially in the case of PLs. The abundance of PLs is roughly similar across both SNT and SB fractions at low solids. However, as solids increase, the PLs decrease in the SNT while proportionally increasing in the SB, indicating that these enzymes may be overexpressed at low solids but only become associated with substrate fractions at higher solids when presumably more binding area is available (Fig. 2C).

GHs – considered to be the workhorses of carbohydrate depolymerization, were the most abundant enzyme class observed across all fractions but exhibited only a <1.5-fold increase in their total abundance while solids increased by 5-fold (Fig. 2C). Proteins from 53 GH families were identified and had varying trends with solids loadings across the three fractions (Supplementary Fig. 10, Supplementary Data 7). To tease apart the overall GH trends observed in Fig. 2C, the identified GH proteins were grouped into broad deconstruction categories based on their most common activity, and quantitatively assessed (Fig. 2D, Supplementary Data 6). The contributions from the main chain polysaccharide-hydrolyzing GHs (longer polymer; colored brown in (Fig. 2D) decreased with increased solids in all three fractions. This trend was countered by a noticeable increase in either the small oligosaccharide acting GHs (SNT; orange) or the hemicellulose debranching GHs (PC and SB; purple) (Fig. 2D). These observations indicate that the microbiome adjusts its GH machinery to continue solubilization at high solids with minimal impact to aggregate GH abundance and prompted us to further evaluate functional CAZyme changes across the major component polymers of switchgrass.

**Oligosaccharide acting β-glucosidases increase at higher solids**. Multiple GH proteins across the three main cellulolytic categories (endoglucanases, exoglucanases, and β-glucosidases) were present in the cultures at varying levels. Both endoglucanases (mostly GH9 and GH5, in accordance with other studies[46,47]) and β-glucosidases (primarily GH3) were highly represented, indicating the importance of these cellulolytic activities in the system (Fig. 3, Supplementary Data 7). Unexpectedly, at 150 g/L solids—representing a 5-fold increase in switchgrass—the collective abundance of endoglucanases, a main constituent of many cellulose degrading microbes, did not change in the SB fraction, decreased significantly in the SNT fraction, and only increased moderately (1.4-fold in 120 g/L) in the PC fraction relative to 30 g/L (Fig. 3, Supplementary Fig. 11, 12D). Also, exoglucanases- considered the main enzymes responsible for cellulose chain depolymerization, exhibited a similar abundance trend with increasing solids,

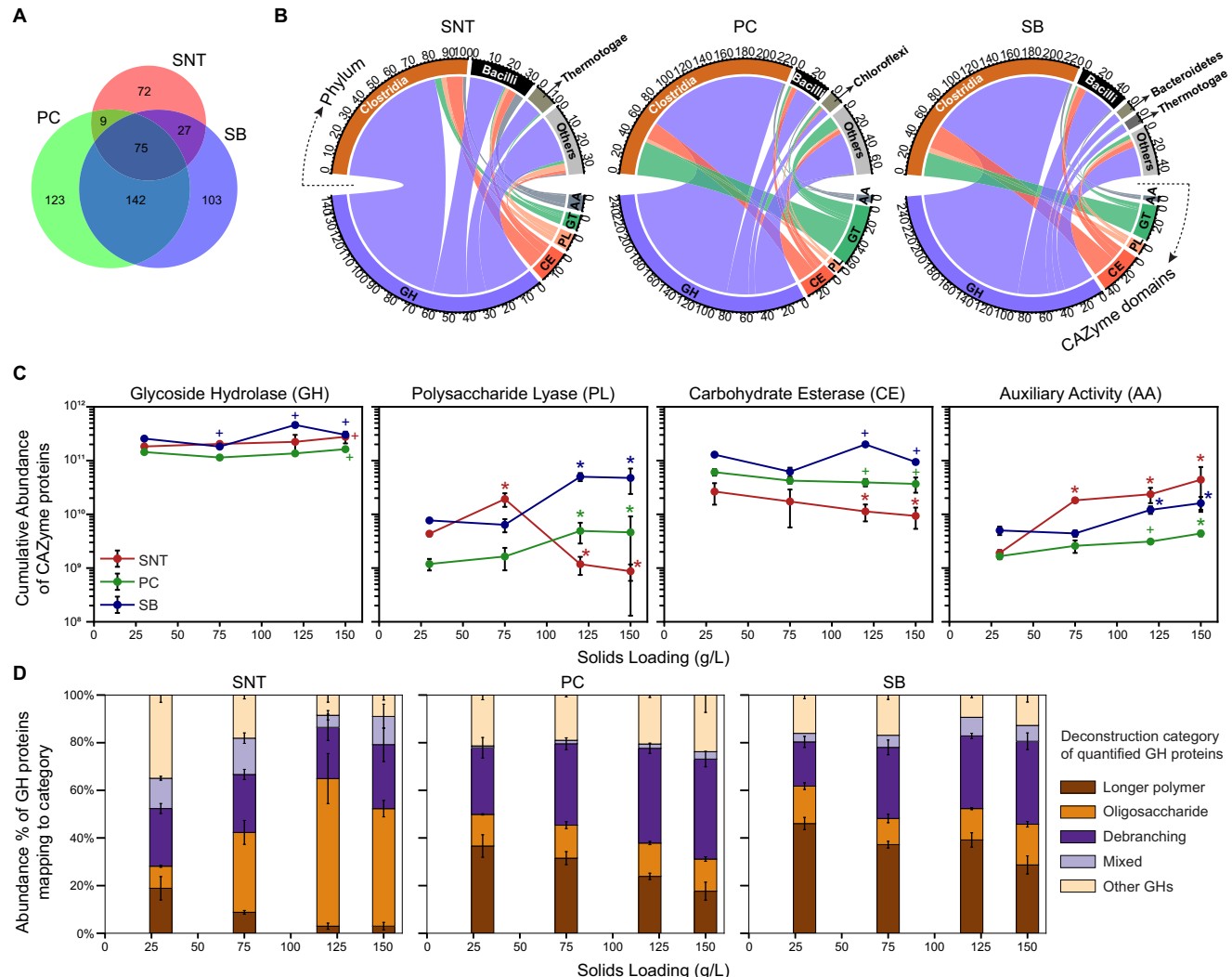

**Fig. 2 Overview of the measured CAZymes' repertoire in the three fractions. A** Venn diagram showing the overlap of CAZymes quantified in the three fractions. **B** For each fraction, the distribution of quantified CAZymes across the CAZyme classes (GH, CE, PL, GT, AA) with linkages showing the contributing taxonomic phyla (top half). Phyla producing <10 CAZymes were grouped as 'Others'. **C** Aggregate abundance trend of CAZymes from different CAZyme classes with increasing solids loadings per fraction. A two-tailed Welch's t-test was performed with Benjamini Hochberg FDR correction from each solids loading versus the 30 g/L loading. + means adjusted (adj.) p-value ≤ 0.05 and * means adj. p-value ≤ 0.05 with absolute fold change ≥2× vs. respective 30 g/L condition. Exact p-values for each comparison are listed in Supplementary Data 6. **D** The relative distribution of each deconstruction sub-category within GHs. Data are presented as mean values ± SD (n = 4 for 30 and 120 g/L, n = 3 for 75 g/L, and n = 5 for 150 g/L). SNT supernatant, PC planktonic cells, SB substrate bound, GH glycoside hydrolase, CE carbohydrate esterase, PL polysaccharide lyase, GT glycosyl transferase, AA auxiliary activity. Source data are provided as a Source Data file.

specifically with regard to SB and SNT fractions. In contrast, a significant increase in the cumulative abundance of β-glucosidases was observed especially in the SNT fraction (~12-fold increase) where small oligosaccharides such as cellobiose might occur (Figs. 2D, 3, Supplementary Fig. 11, 12A). β-glucosidases are known to relieve end-product inhibition of cellulases, improve conversion yields and cellulose saccharification rates, and lower total enzyme requirements[51,52]. These observations are consistent with a synergistic role of β-glucosidases in maintaining cellulose hydrolysis at higher solids loadings. Taxa-level dynamics of β-glucosidase expression were also observed (Supplementary Fig. 12). While members of multiple phyla produced β-glucosidases, the increase in abundance at high solids was driven by the phylum *Dictyoglomi* (Supplementary Fig. 12A). Distantly related to *Caldicellulosiruptor* spp., members of this phylum are known to be extremely thermophilic, possess rigid cell membranes, and produce thermostable enzymes[53–55].

**Hemicellulose debranching enzymes increase at higher solids.** Multiple proteins across different GH and CE families were grouped into a variety of hemicellulolytic activities and their trends with increasing solids loading explored. These activities included: xylanases, xylosidases, arabinosidases, glucuronidases, fucosidases, galactosidases, mannosidases, and acetyl-xylan esterases (Fig. 3, Supplementary Fig. 11, Supplementary Data 7). Both the major xylan-hydrolyzing enzymatic classes, xylanases and xylosidases, were represented by numerous proteins and were most prevalent in the SB fraction (Fig. 3, Supplementary Fig. 12b, Supplementary Data 7). The total abundance of xylanases at 150 g/L solids loading decreased significantly in the SNT and PC fractions (~10-fold and ~2-fold, respectively) compared to 30 g/L, but remained relatively unchanged in the SB fraction (Fig. 3, Supplementary Fig. 12B). However, xylosidases exhibited a consistent increase at higher solids in the SB fraction (~3-fold in 120 g/L and 1.9-fold in 150 g/L) (Fig. 3, Supplementary Fig. 12B).

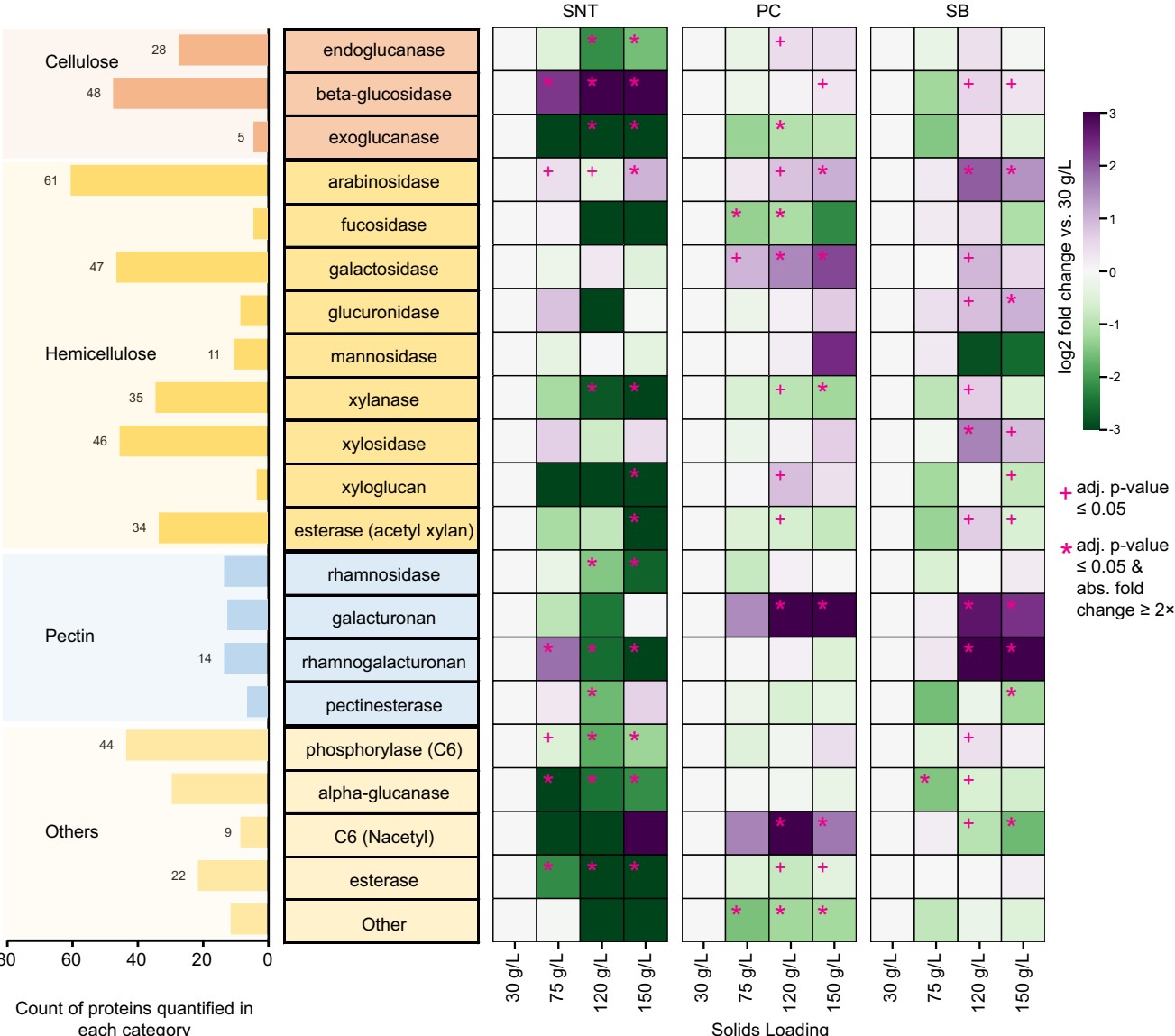

**Fig. 3 Analysis of CAZyme functional categories at higher solids loadings.** (left side) Enumeration of quantified unique CAZyme proteins across different CAZyme families (GH, CE, PL) after organizing them based on functional annotation as described in Supplementary Data 3. (right side) Heatmap depicting the change in aggregate abundance of CAZymes in each functional category across solids loadings for each fraction. Log$_2$ differences from 30 g/L solids in the respective fraction are shown. A two-tailed Welch's $t$-test was performed with Benjamini Hochberg FDR correction for each solids loading versus the 30 g/L loading. + means adjusted (adj.) $p$-value ≤ 0.05 and * means adj. $p$-value ≤ 0.05 with absolute fold change ≥2× vs. respective 30 g/L condition. Exact $p$-values for each comparison are listed in Supplementary Data 6. SNT: supernatant, PC: planktonic cells, and SB: substrate bound fraction. Source data are provided as a Source Data file.

Since xylan breakdown was undiminished at high solids loading, the observations here suggest xylosidases may be needed for maintaining overall xylan hydrolysis—similar to what was observed with β-glucosidases. Likewise, the importance of xylosidases at higher solids was further substantiated by community dynamics, where the increase was driven by a protein group from *Bacteroidetes* (Supplementary Fig. 12B), members of whom are key xylanolytic organisms in the rumen and human gut ecosystems[56].

Along with xylan-acting enzymes, numerous hemicellulose-acting and debranching enzymes were expressed by the microbial community (Fig. 3, Supplementary Data 7). In particular, proteins with arabinosidase activity were highly represented (61 proteins) and demonstrated increased abundance with increasing solids loadings in all fractions (Fig. 3, Supplementary Fig. 12E). Even though arabinose makes up only ~4% of switchgrass carbohydrate content[44,57], the disproportionately high number of proteins dedicated to cleaving branching arabinose residues or arabinan, coupled with their uniform increase in abundance, suggests a need for arabinosidases at higher solids. Similarly, despite galactose constituting <2% of the sugar content in switchgrass[57], 47 proteins with galactosidase activity were detected. This group is composed primarily of family GH2 and contains CAZymes with both α- and β-galactosidase activities that cleave the branching galactose residues in mannan and xylan. Their abundance significantly increased in the PC fraction at higher solids (Fig. 3), a trend driven by the phylum *Chloroflexi* along with *Firmicutes-Clostridia* (Supplementary Fig 12F). The filamentous *Chloroflexi* are enriched in the PC and SB fractions and are known

carbohydrate fermenters. They are likely involved in the degradation of polymeric organic compounds facilitating the growth of other bacteria[58] to maintain overall solubilization. Conversely, the abundance of other enzymes considered important for solubilization, such as acetyl-xylan esterases (34 protein groups) and $C_6$ phosphorylases (44 protein groups including cellobiose phosphorylases), decreased in abundance or remained relatively unchanged at higher solids loadings although they were well-represented (Fig. 3, Supplementary Data 7).

**Pectin-acting enzymes increase in substrate-associated fractions.** Pectin represents a small fraction of switchgrass on a mass basis (~2%), but it provides structural tenacity and contributes to its recalcitrance[59,60]. Numerous pectin-acting proteins representing different CAZyme families (PL, GH, and CE) were expressed by the microbiome, which can be roughly grouped into four functional categories: galacturonan lyase, rhamnogalacturonan

lyase, rhamnosidase, and pectinesterase (Fig. 3, Supplementary Fig. 11, Supplementary Data 7). While pectinesterases and rhamnosidases had decreasing abundances at higher solids, significant increases in the two pectin lyases, driven by the phyla *Firmicutes*, were observed in PC and/or SB fractions at high solids. These pectin lyases can be particularly important because they cleave the alpha-1,4- linkages in esterified pectin (main chain) without requiring accessory enzymes or water molecules, as is the case with PLs[60].

**Bacterial AA6 proteins increase dramatically at higher solids.** Auxiliary activities (AAs) are a recently constituted class of CAZymes that act on lignin or are involved in modification of polysaccharides such as cellulose in the plant cell wall[61]. A significant increase in the abundance of AA enzymes was observed across all three fractions as solids loadings increased (Fig. 2C). In the current study, 3 AA families were identified (Fig. 4A). AA2 (peroxidase) and AA3 (glucose-methanol-choline (GMC)

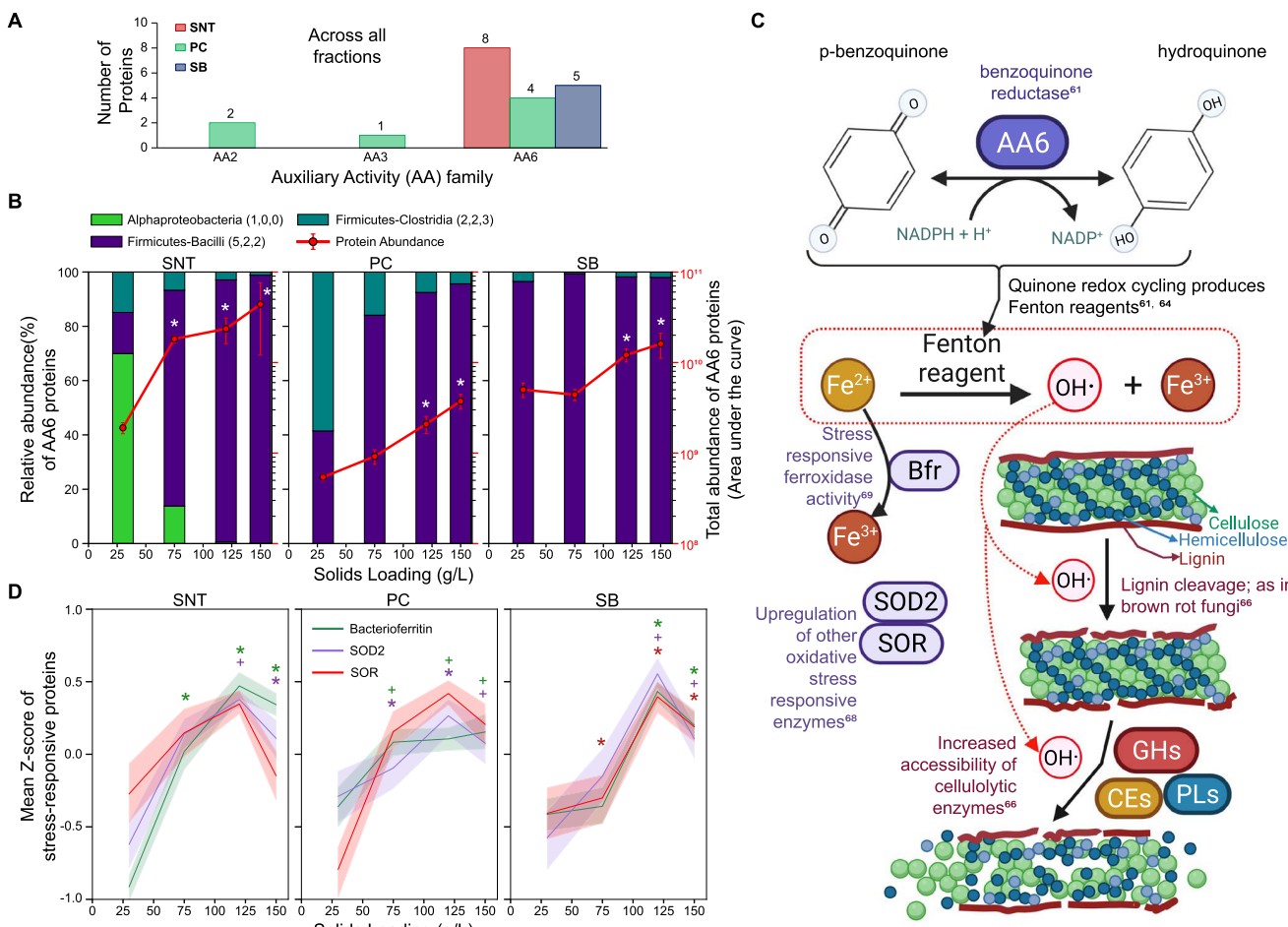

**Fig. 4 The abundance of bacterial AA6 proteins as a function of higher solids loadings. A** The number of proteins expressed from the different auxiliary activity (AA) families in each fraction. **B** The change in total abundance of AA6 proteins across solids loadings per fraction (right Y-axis, line plot) with relative abundance contributions by phyla (left Y-axis, bar plot). Error bars are ± standard deviation and * means two-tailed Welch's t-test corrected p-value < 0.05 and absolute fold change ≥2× versus respective 30 g/L condition. **C** A proposed mechanistic hypothesis based on observed increases in protein abundances and known functions depicting the suggested role of AA6 proteins in enabling solubilization at higher solids loadings via ROS and Fenton chemistry. **D** The mean trends of proteins that are known to respond to oxidative stress across solids loadings per fraction. Bacterioferritin (Bfr; 18, 23, 20 proteins in SNT, PC, SB respectively), superoxide dismutase (SOD2;15,11,7) and superoxide reductase (SOR; 9, 10, 13). Mean Z-scores are shown with 50% confidence interval. + means two-tailed Welch's t-test p-value < 0.05 and * means p-value < 0.05 and absolute fold change ≥2× versus respective 30 g/L condition. Exact p-values for each comparison are listed in Supplementary Data 9. SNT: supernatant, PC: planktonic cells, and SB: substrate bound fraction. Source data are provided as a Source Data file.

oxidoreductase) were identified only in the PC fraction at relatively low abundance. The cumulative abundance of AA2 proteins had a decreasing trend with increasing solids loadings (Supplementary Fig. 13A). AA3 was an order of magnitude less abundant than AA2 and present only at higher solids but was identified by only one peptide (Supplementary Fig. 13B). In contrast, AA6 proteins were 1-2 orders of magnitude more abundant than AA2 and represented by multiple proteins (11/14 AA proteins mapping to family AA6) across the three fractions. AA6s exhibited significantly increased abundance at higher solids (~23-fold in SNT at 150 g/L); a trend that was consistently observed across all three sample fractions (Fig. 4A, B). Members of the phyla *Firmicutes-Clostridia* and *Firmicutes-Bacilli* were the primary contributors to the AA6 protein group, but the increase in abundance at higher solids was driven by the phylum *Firmicutes-Bacilli* (Fig. 4B). AA6 family enzymes are characterized as 1,4-benzoquinone reductases and catalyze the NADPH-mediated conversion *p*-benzoquinone to hydroquinone[61] (Fig. 4C). Although the switchgrass was not pretreated in the current study, *p*-benzoquinone is a pretreatment- and lignin-derived growth/enzyme inhibitor that is difficult to remove, suggesting that AA6-like enzymes could be involved in biomass hydrolysate detoxification[62,63]. While recent studies have identified an oxidoreductase gene, *ZMO1116*, that can convert benzoquinone to non-toxic hydroquinone[62,63], multiple sequence alignment analysis of the AA6 proteins identified here only revealed local sequence similarities to ZMO1116 (Supplementary Fig. 14), suggesting a potentially different mode of action in this system.

Similar to biochemical mechanisms observed in brown-rot fungi, AA6 in the bacterial community may also be involved in the production of extracellular oxyradicals for lignin modification. AA6, or quinone reductase in brown-rot fungi, is involved in the generation of extracellular Fenton reagents via redox cycling[61,64]. Unlike white-rot fungi, which degrade and utilize lignin, cellulolytic brown-rot fungi modify lignin using non-enzymatic, energetically inexpensive Fenton chemistry to breach the lignin barrier and enable enzyme access to the underlying polysaccharides[65,66]. Fenton reactions make use of peroxide ($H_2O_2$) and iron ($Fe^{2+}$ to $Fe^{3+}$) to produce reactive oxygen species (ROS), such as highly reactive hydroxyl radicals, which can non-specifically cleave lignin[66]. Since AA6 in the CAZy database (all annotated AA6 are from fungi) are suspected to produce Fenton reagents[61], the bacterial community may follow a route similar to that of brown-rot fungi to effectively solubilize increasing concentrations of switchgrass (Fig. 4C). Although proteins annotated for direct production of $H_2O_2$ were not identified in the samples, multiple oxidases were observed (Supplementary Data 8). Additionally, hallmarks of oxidative stress such as superoxide dismutases[67,68] (SOD2) and superoxide reductases[68] (SOR) were also present in the samples, both of which increased significantly in abundance with increasing solids in one (SOR, in SB fraction) or all (SOD2) fractions (Fig. 4C, D, Supplementary Fig. 13C, D, Supplementary Data 9). Furthermore, bacterioferritins from multiple taxa were also detected and significantly increased in all three fractions (Fig. 4d, Supplementary Fig. 13C). Bacterioferritins are enzymes with ferroxidase activity; ferroxidases are upregulated in brown-rot fungi during Fenton reactions[69]. Additionally, as is the case in brown-rot fungi, limited proteins for lignin metabolism/assimilation were observed in the samples, indicating lignin cleavage may enhance enzyme accessibility rather than be used for energy (Supplementary Fig. 15–20). Notably and in contrast to fungal AA6s which are known to act aerobically, these bacterial AA6s appear to have an anoxic mode of operation, which has also been observed in a recent study[70]. Specifically, these results suggest that in bacterial communities, AA6 enzymes (here produced by *Firmicutes*) could

be involved in the generation of Fenton reagents and subsequent cleavage of the lignin barrier to increase the accessibility of lignocellulose to attack by CAZymes.

## Discussion

The thermophilic, lignocellulose-fermenting, methanogenic anaerobic microbiome reported here exhibits a key feature desired for an industrial process: undiminished fractional carbohydrate solubilization with increasing substrate loading (Fig. 1). In search of clues as to how this is achieved, this study reports the first metaproteomic characterization of a microbiome as a function of substrate loading across different fractions, focusing on salient CAZyme and methanogenesis features. Comparing relative protein abundances at 150 g/L and 30 g/L substrate loadings in Fig. 5A, a greater than 2-fold increase was observed in one or more fractions for β-glucosidases, hemicellulose-debranching and xylosidase enzymes, pectinases, and AA6 enzymes, as well as all hydrogenotrophic methanogenesis pathway enzymes. The latter corroborates the observed increase in methane production as solids increased (Fig. 1C), and as described before in anaerobic thermophilic microbiomes[71], is the preferred methanogenesis route under thermophilic temperatures[72,73]. Coupled with syntrophic acetate oxidation for acetate degradation, this metabolic process is performed by members of *Euryarchaeota*, who may not need to be proximal to lignocellulosic substrate. As can be seen from Fig. 5B, this process primarily occurred in the cell containing fractions, PC and SB, with the bulk of the activity in the PC fraction for 150 g/L.

Considering enzymatic deconstruction of lignocellulose, the largest increases were seen for β-glucosidases in the SNT, pectinases in the PC and SB fractions, and for AA6 family, debranching, and xylosidase enzymes in all fractions. The observation that the microbiome produces these particular enzymes instead of increasing amounts of backbone depolymerizing enzymes (such as endoglucanases and xylanases) suggests their importance for undiminished hydrolysis at high solids—potentially through removal of accumulated inhibitive (hemi) cellulose solubilization products (β-glucosidases and xylosidases) and enabling enzymatic access to the preferred carbohydrate polymers. For example, debranching of xylan "decorations" alters its interaction with other cell-wall polymers (cellulose and lignin) effectively decreasing the recalcitrance of feedstock[74]. Pectinases/polygalacturonases synergistically improve the hydrolytic efficiency of cellulases by removing pectin, and thereby improve access to cellulose[75–77]. Finally, the marked increase in AA6 enzymes—purported to utilize Fenton chemistry for deconstruction and permeation of the lignin barrier—fits this paradigm. These AA6 family enzymes are also particularly notable since oxidative reactions are not generally associated with anaerobic lignocellulose fermentation. However, Schalk et al.[43] recently documented a role for fungal driven AA6-mediated Fenton chemistry in the termite gut. Although further investigation is needed in the mechanism proposed here, the observations suggest that AA6 plays an important synergistic role with other CAZyme categories for high solids deconstruction under anaerobic thermophilic conditions. Our results also provide insight into the spatial location of bacterial enzymes, their phylogenetic origin(s), and how these are impacted by substrate loading (Fig. 5B). At the substrate loading of 30 g/L, β-glucosidases and AA6 enzymes are relatively more prominent in the substrate-bound fraction, but at 150 g/L become most prominent in the supernatant. In contrast, hemicellulose-debranching enzymes and pectinases are most prominent in the supernatant at 30 g/L, but in the substrate-bound fraction at 150 g/L. The clear difference in functional distribution across fractions highlights the importance of this

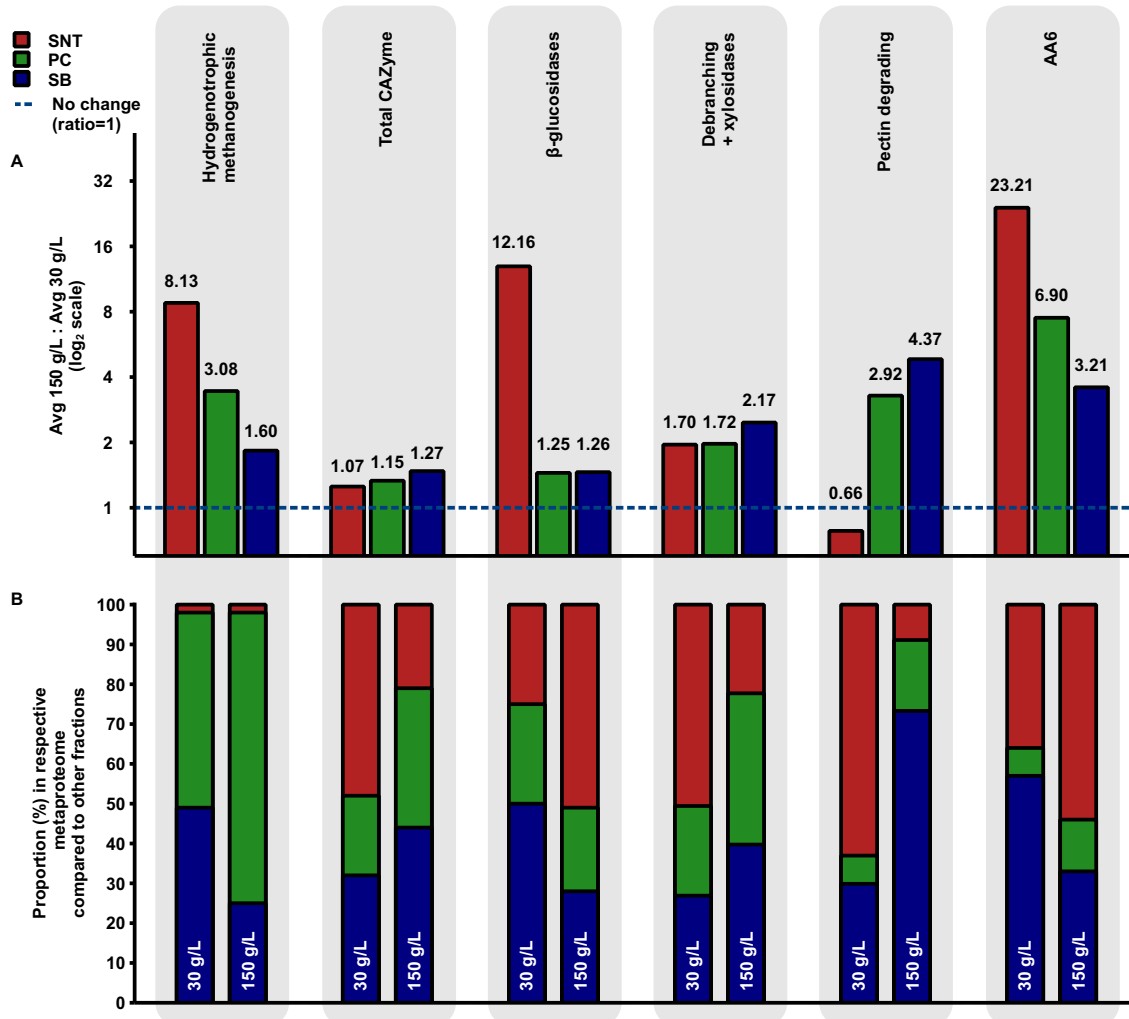

**Fig. 5 Microbiome functionally and spatially orchestrates enzyme expression to continue solubilization at high solids. A** Average fold changes (ratios) in aggregate protein abundance of the major enzymatic categories at 150 g/L solids loading relative to 30 g/L solids loading compared across the three fractions. **B** The relative proportion of the respective fraction metaproteome that each of these enzymatic categories make up is shown here. This qualitative representation depicts the spatial distribution of these enzymatic categories among the three fractions and changes in their distribution from 30 g/L to 150 g/L. SNT: supernatant, PC: planktonic cells, and SB: substrate bound fraction. Source data are provided as a Source Data file.

approach to provide both relevant spatial location information as well as enhancements to the depth of the metaproteomic measurement.

With increasing solids, the collective protein synthesis effort of the microbiome essentially shifts toward enzymes that act on compounds known to be inhibitors of deconstruction - cellobiose[78–80], hemicellulose[81,82], pectin[83,84], and lignin[85–87], with roughly constant effort devoted to the CAZymes that mediate mainline (hemi)cellulolytic deconstruction. These observations provided by metaproteomics provide an unprecedented level of molecular detail which will be critical to inform development of defined cultures for conversion of cellulosic biomass to products other than methane, which thus far exhibit decreasing fractional carbohydrate solubilization with increasing substrate loading[20,21,88].

## Methods

**Feedstocks and microbiome origin.** Mid-season switchgrass (*Panicum virgatum* L., Cave in Rock) harvested in June 2012 at Rock Springs Research Farm (Spring Mills, PA, 40.71040290971864°N, −77.94974506560891 °W) was used as the substrate in this study. Characterization, preparation and storage information is described in detail in Liang et al.[44]. An existing microbial community was used in this study, as described in Liang et al.[44]. The inoculum for the initial enrichment

was obtained by sampling the anaerobic digester from Vermont Technical College (Randolph, Vermont). The enrichment was matured as a microbiome for over 120 days before experiments with different residence times were initiated, the results of which are described in Liang et al.[44]. Reactor HR1 (formerly named R2 in Liang et al.[44]) ran at various residence times (RT) and was cultivated for a total 214 days at 30 g/L solids loading before it was transitioned to 75 g/L to generate the additional higher solids data for the study described here. A detailed characterization of the microbiome can be found in Liang et al.[44]. Potential microbial contributions via the addition of unsterilized feedstock to the microbiome were investigated by Liang et al.[44], and found to be small.

**Cultivation medium.** Mixed enrichment (ME) medium was used in this study[44], except that the amount of Wolfe's vitamin solution was doubled for 75 g/L and up compared to 30 g/L. The following stock solutions were used: Wolfe's modified elixir stock solution[89] (50×), Wolfe's vitamin solution[89] (50×), Ammonium & Phosphate stock solution (50×) and Ferrous iron stock solution (1000×) are described in Liang et al.[44]. The solutions were either autoclaved (Ferrous iron stock solution) or filter-sterilized (all other stock solutions). Switchgrass was added (without sterilization) to the reactor to a concentration of 30, 75, 120, or 150 g/L (as-is basis with 6.05% moisture) according to the solids loading per feeding period. The pH of the medium was adjusted with 1 N Sodium Hydroxide solution after feeding to maintain a value above 6.0 as needed.

**Cultivation system.** The bioreactor set-up used for this study was a Qplus multi-bioreactor system (Sartorius Stedim, Bohemia NY) which includes Module Operator Service Program (MFCS) data collection software (3.0, level 43, 2008

Sartorius Stedim Systems) recording primary fermentation data (pH, temperature, stirrer rpm, pH controlled base addition). The scientific plotting package Veusz was used for visualization of primary fermentation and analyzed data (v3.4.01, https://veusz.github.io/). Additional details for the bioreactor set-up are described in Liang et al.[44]. The 1-liter working volume bioreactor was stirred at 280 rpm with the temperature controlled at 55 °C. The set up was operated semi-continuously at a residence time of 10 days; every day 10% of working volume (100 mL) of the broth was removed and replaced by fresh switchgrass and media components totaling 100 mL in volume. The removed broth was used for analysis. When the solids loading in the feed was equal to 150 g/L, the stirring speed for HR1 was increased to 500 rpm for one-minute right after the sampling and feeding event to ensure efficient mixing. Feeding aliquots of 100 mL of sterilized growth medium + switchgrass with pH of 8.0 at room temperature was sufficient to keep the pH in the bioreactor above 6.0 for all residence times.

**Cultivation start-up and operation.** As described in Liang et al.[44], prior to this study the bioreactor was operated semi-continuously for 214 days, fed with medium containing 30 g/L switchgrass at various residence times (20–3.3 days). For an RT of 10 days and 30 g/L solids loading the total cultivation time was 50 days of operation as described in Liang et al.[44] and fundamental data and samples for subsequent analysis for that solids loading were incorporated into the results described here. For the current study the reactors were fed with medium containing increasing loadings of switchgrass starting at 75 g/L, 120 g/L and 150 g/L. At the start of this study the solids loading was increased from 30 to 75 g/L and the RT fixed at 10 days (from 3.3 days). The frequency of slurry removal and feeding was maintained at 10 times per RT resulting in 1 feeding-sampling event each day for the 10-day residence time and the length of the overall cultivation time.

The microbiome was operated for 506 days after transition to 75 g/L and RT = 10 days. It was operated for 204 days at 75 g/L switchgrass loading in the feed and was then fed 120 g/L switchgrass for 85 days. Finally, the solids loading was increased to 150 g/L and maintained at this level for 217 days. The reactor went through medium component optimization during the first 150 days at 75 g/L solids loading and RT = 10 days and was fed with the medium described earlier from 151th day on. Steady state at this condition was considered to start from 161th day. For all the other conditions, the reactor was considered to have reached steady state after 3 RTs. Total reactor operating length and steady state length at each condition is summarized in Supplementary Table 1 and in Supplementary Fig. 1. Slurry was withdrawn via a 50 mL pipette by removing a plug from the head-plate as described in Liang et al.[44]. When the solids loading was 150 g/L, the opening of the 50 mL pipette was cut wider to enable sample removal, the pipetted-volume indication was adjusted accordingly (total volume equaling 50 mL).

**Biogas measurement.** Measurement of biogas production rate was conducted using a wet tip gas meter (www.wettipgasmeter.com) filled with acidified water (pH < 2). At each feeding, data was recorded either manually or via a data logger (HOBO Pendant Event, 64 K, Onset Computer Corporation, Pocasset, MA). The concentrations of $CH_4$, $CO_2$, $H_2$ and $N_2$ were measured by a Model 310 Education Gas Chromatograph (SRI Instruments, Torrance, CA and a 1.8 m x 3 mm stainless steel HayeSep D packed column at 50 °C) with a thermal conductivity detector (150 °C) using helium (20 mL/min) for $CH_4$, $CO_2$ and $N_2$ and nitrogen (13.5 mL/min) for $H_2$ as carrier gases. For data see Supplementary Table 2 and Supplementary Fig. 1).

**Fractional carbohydrate solubilization.** Measurement of fractional carbohydrate solubilization, FCS, is based on the amount of glucose, xylose and arabinose in the solids before and after fermentation. Sugar content is determined by acid hydrolysis via the quantitative saccharification protocol[90,91]. Whole broth samples were centrifuged for 10–11 min at $2,800 \times g$ (30–120 g/L) or $12,000 \times g$ (150 g/L). Pelleted solids were dried overnight in a 60 °C oven (30–120 g/L) or by lyophilization (150 g/L), weighed and then subjected to quantitative saccharification. The hydrolysis products monomeric glucose, xylose and arabinose were then quantified by HPLC (Waters, Milford, MA) with an Aminex HPX-87H column (Bio-Rad, Hercules, CA) at 60 °C and detected by refractive index. HPLC eluent was 5 mM sulfuric acid with a flow rate of 0.6 mL/min. FCS equals to the mass of the initial carbohydrate minus the mass of the final carbohydrate divided by the initial carbohydrate[44]. Steady state FCS data as shown in Fig. 1b is based on averages of two or more samples after at least 3 residence times following a change in solids loading[90]. For data see Supplementary Table 2 and Supplementary Fig. 1).

**Quantification of volatile fatty acids.** Measurement of volatile fatty acids (VFA) was done by analyzing filtered liquid broth samples for formic acid, acetic acid, propionic acid, butyric acid, iso-butyric acid and valeric acid. All measurements were performed in duplicate against a known standard (Volatile-free acid mix standard, 46975-U SUPELCO, Sigma-Aldrich). Analysis was by HPLC with an Aminex HPX-87H column as described for Fractional Carbohydrate Solubilization. For data see Supplementary Table 2 and Supplementary Fig. 1).

**Assembly of the bioreactor gene catalog for metaproteomics analysis.** The metagenomic data generated in Liang et al.[44], which described the development of a stable switchgrass-fermenting microbiome at various residence times and temperatures, was examined for functional gene profile diversity. Metagenomic

sequence reads of 15 samples were downloaded from the US Department of Energy's Joint Genome Institute website under proposal ID 502908 (https://genome.jgi.doe.gov/portal/ChaofEnCultures/ChaofEnCultures.info.html). All metagenomic sequence files were combined into a single file in a conglomerative format to capture as many prokaryote coding regions as possible. Quality filtering and trimming was performed using Atropos with a minimum sequence length of 50 nucleotides and a minimum Phred score of 30 to ensure only the highest quality sequences remained. The high-quality sequences were then assembled into 506,732 contigs (a total of 777,641,587 bp) using MEGAHIT and prokaryote genes identified using Prodigal. The minimum contig size was 200 bp and the maximum contig was 606,950 bp with an average of 1,535 bp in length and the N50 of 6,420 bp. In all, 1,076,153 prokaryote genes were identified from these contigs. The genes were then assembled into a bioreactor gene catalog in FASTA format for use in metaproteomics analysis. The bioreactor gene set comprised of 485,820 full length prokaryote genes and 590,333 fragmented prokaryote genes. For metaproteomic analysis, only the full-length genes were kept for inclusion in the bioreactor gene atlas to ensure high quality results.

**Sample fractionation, preparation, and measurements for 2D-LC-MS/MS-based metaproteomics.** Switchgrass fermentation whole-broth samples for each of the four solids loadings (30 g/L, $n = 4$; 75 g/L, $n = 3$; 120 g/L, $n = 4$; and 150 g/L, $n = 5$) at steady-state conditions were collected and stored at −80 °C for analysis. For metaproteomic measurements, samples were thawed under cold running water to keep the temperature at 4 °C. Each sample was then fractionated into three phases by centrifugation to enrich for the microbes and enzymes adhered to the lignocellulosic substrate (or substrate bound fraction- "SB"), the planktonic microbes (or planktonic cells fraction- "PC"), and proteins which were secreted (supernatant fraction- "SNT") as described. For centrifugation-mediated enrichment, each sample was centrifuged at $200 \times g$ for 10 min to pellet the residual solid substrates and bound microbial matter (SB fraction), and the resulting pre-supernatant primarily containing the free cells and enzymes was transferred into a new tube. This pre-supernatant was further centrifuged at $1000 \times g$ for 20 min. The resulting pellet was the PC fraction, enriched for unbound microbial cells and the resulting supernatant was the SNT fraction, enriched for free proteins. Each segregated fraction was then analyzed separately for metaproteomics.

Each sample combination of loading and fraction was lysed by bead beating with 0.15 mm zirconium oxide beads in Tris-HCl (100 mM at pH 8.0) containing 4% SDS (sodium dodecyl sulfate, Sigma) and 10 mM DL-Dithiothreitol (Sigma). Resulting lysates were precleared by centrifugation at $21,000 \times g$ for 10 min, incubated at 90 °C for 10 min to denature proteins, and adjusted to 30 mM IAA (iodoacetamide, Sigma) followed by a 20-minute incubation in darkness at room temperature to alkylate/block cysteine residues. Crude protein was isolated and cleaned up by chloroform-methanol-extraction. Resulting protein pellets were washed with methanol, air dried, and re-solubilized in freshly prepared ABC (ammonium bicarbonate, Sigma) buffer (100 mM, pH 8.0) containing 4% SDC (sodium deoxycholate, Sigma). Protein concentrations were determined using the BCA (Bicinchoninic Acid) protein assay (Pierce). Fixed amount of protein sample (300 µg) was concentrated on a 10 kDa MWCO centrifugal concentrator (Vivaspin500 PES; Sartorius), rinsed with ABC buffer, and digested in situ with MS grade trypsin protease (1:75 w/w; Pierce-Thermo Scientific) overnight at room temperature, and again for 3 h after fresh trypsin addition. After digestion, samples were filtered through the concentrator membrane to remove under-digested proteins and collect tryptic peptides. The resulting peptide solution was acidified with formic acid (FA; LC/MS grade) to a final concentration of 1% to precipitate remaining SDC, followed by additional removal of the precipitate using water-saturated ethyl-acetate. Peptides were then concentrated to dryness using a SpeedVac, resuspended in 0.5% formic acid, quantified by BCA protein assay, and analyzed by 2D-LC-MS/MS using a Vanquish UHPLC system (Thermo Scientific) with autosampler coupled to an Orbitrap Q Exactive Plus mass spectrometer (Thermo Scientific).

Fixed amounts of peptides depending on the complexity of the fraction: 10 µg peptides for SB, 14 µg for PC, and 6 µg for SNT fractions, respectively, were loaded into an in-house built triphasic MudPIT back column (100 µm ID packed with RP-SCX-RP; RP- reversed-phase $C_{18}$ resin, 5 µm Kinetex, Phenomenex; SCX- strong-cation exchange, 5 µm Luna) coupled to an in-house pulled nanospray emitter (75 µm ID) packed with 30 cm of 5 µm RP $C_{18}$ resin for online 2D HPLC separation. Peptides from each sample type were then trapped, desalted, separated, and analyzed over successive salt cuts of ammonium acetate (35, 50, 100, and 500 mM), each followed by an organic gradient to elute peptides. The eluting peptides were measured and sequenced by the mass spectrometer (operated via Xcalibur v.4.2.47, Thermo Scientific) in data-dependent mode.

**Metaproteomics data analysis.** The acquired peptide fragmentation spectra were searched against the metaproteome database generated from the 30 g/L samples' metagenomes (as described above) appended with common contaminant proteins, and the switchgrass (*Panicum virgatum*) proteome employing a target decoy approach using the MS Amanda algorithm (v2.0) integrated in Proteome Discoverer software (version 2.3.0.523, Thermo Scientific). The resulting peptide spectrum matches (PSMs) were required to be at least 5 amino acids long, fully tryptic with a maximum 2 missed cleavages, contain a static modification of 57.0214 Da on cysteine (carbamidomethylated) residues, and a dynamic modification of 15.9949 Da on methionine (oxidized) residues. False-discovery rates (FDRs) were controlled at 1% (IMP-Elutator node in Proteome Discoverer) at both

the PSM and peptide levels. Peptides were quantified by extracting the chromatographic area-under-the-curve (AUC) and match between runs was conducted by performing a grouped consensus step in Proteome Discoverer per fraction. To bypass ambiguities related to shared peptides among very similar to identical proteins, the proteins with peptide evidence were grouped based on sequence homology and peptide evidence by enabling protein grouping in Proteome Discoverer. The peptides were then assigned to protein groups (represented by master seed proteins) utilizing the inbuilt parsimony principle. The AUCs of peptides uniquely mapping to a protein group were summed to obtain protein (group) abundances and protein groups with at least 1 unique peptide were considered for further analysis. The resulting protein lists were filtered to select highly confident protein groups (FDR of ≤1%) which had at least 2 MS/MS spectra captured, were detected in >2 samples across the fraction experimental set. The resulting protein abundances were bioinformatically processed as described previously[92]. Briefly, protein (group) abundances in each fraction were log$_2$ transformed and distributions were normalized (LOESS normalization followed by median centering) across samples using InfernoRDN. Missing values were imputed to simulate the mass spectrometer's limit of detection using mean minus 2.2 times the standard deviation with a width of 0.3 times the standard deviation in Perseus. For each fraction, significant differences in protein abundances between the 30 g/L condition and the higher solids loadings were assessed by two-tailed $t$-test at a Benjamini-Hochberg FDR corrected $p$-value of ≤0.05.

For functional analysis, the metaproteome database was functionally annotated with Kyoto Encyclopedia of Genes and Genomes (KEGG) orthologous (KO) terms, E.C. numbers, and definitions as obtained from GhostKOALA[93], eggNOG-mapper[94], and KEGG GHOSTX searches. Protein-level phylum level taxonomical assignments were assigned to each protein group using GhostKOALA after implementing a cutoff of ≥100 score. Additional taxonomic information as needed for select proteins was inferred by tblastn analysis against the metagenomic bins described in the previous study for the metagenomes. Additional taxonomic annotations for the metagenomic bins were obtained using CheckM and GTDB-Tk analyses performed in KBase as needed. For identification for CAZymes in the metaproteome, dbCAN2 meta server[49] was run and proteins were qualified as CAZymes based on software recommendations and after manual curation. The dbCAN2 searches were performed using the HMMER, DIAMOND, and Hotpep tools and proteins annotated by ≥2 of these tools were qualified as CAZymes. Sequences qualified as CAZyme by <2 tools were manually inspected. HMMER annotations took priority over DIAMOND and Hotpep tools. Additionally, assignment of dockerin domains was done using InterProScan for all the proteins in the metaproteome database. For assessment of proteins related to cellulosomes, the presence of CAZyme domain with a dockerin domain was needed. KEGG and eggNOG assignments for each identified protein group were used to infer the potential function (enzymatic activity) of the proteins. Proteins related to methanogenesis were identified using KEGG and MetaCyc annotations. To examine statistical variance of categories of proteins in each fraction, pairwise Welch's $t$-tests were performed between the respective solids loadings and 30 g/L followed by a Benjamini-Hochberg FDR correction. While comparisons were considered statistically significant if resulting adjusted $p$-value was ≤0.05, in most cases a cut off of 2× fold change was applied to assess significance. Statistics were calculated using Python, in Excel, or in Perseus. All figures were rendered using Python, R, Excel, or BioRender. Python scripting was done using the following libraries: Pandas, NumPy, Seaborn (https://seaborn.pydata.org/), Matplotlib. Venn diagrams were created using BioVenn. Metabolic pathway information was obtained from MetaCyc and KEGG mapper. For sequence alignment of ZMO1116 sequence to the identified AA6 sequences, different multiple sequence alignment tools- MUSCLE, MAFFT, and ClustalOmega were utilized by choosing the protein alignment option with the default parameters.

### 1D-LC-MS/MS metaproteomics measurement and data analysis for estimation of microbial cell density

For estimation of cell density, a volume-based metaproteomics analysis was conducted for steady state aspirates at each of the four solids loading conditions described above. Samples were thawed at 4 °C, mixed, and equal volumes from each sample were processed. Three methodological replicates for each biological replicate were processed. Samples were lysed by bead-beating in Tris-HCl (100 mM at pH 8.0) using 0.15 mm zirconium oxide beads, followed by adjustment to 4% SDS, heat-treatment (95 °C for 10 min), and centrifugation (21,000 g for 10 min). Samples were adjusted to 10 mM DL-Dithiothreitol (10 min at 90 °C) to reduce proteins, cysteines alkylated by 30 mM iodoacetamide (20 min incubation in darkness) and cleaned up via protein aggregation capture[95]. Crude protein was estimated by a Nanodrop OneC spectrophotometer (Thermo Scientific) using 205 nm absorbance. Aggregated protein (on magnetic Sera-Mag (GE Healthcare) beads) was then digested with MS-grade trypsin (fixed amount; Pierce) in 100 mM Tris-HCl, pH 8.0 overnight at 37 °C, and again for 3 h at 37 °C the following day. Tryptic peptides were acidified to 0.5% formic acid, filtered through a 10 kDa MWCO spin filter (Vivaspin 500; Sartorius), and quantified by Nanodrop OneC. Samples were loaded by volume, i.e., 3 μL of peptides from each sample, and analyzed by 1D LC-MS/MS using a Vanquish uHPLC coupled directly to an Orbitrap Q Exactive mass spectrometer (Thermo Scientific), as previously described[92]. Peptides were separated by a 180 min organic gradient across an in-house pulled nanospray emitter (inner diameter, 75 μm)

packed with 15 cm of 1.7-micron Kinetex C$_{18}$ reversed-phase resin (Phenomenex). Eluting peptides were measured and sequenced by data-dependent acquisition. Peptide fragmentation spectra were searched against the concatenated databases and quantified using Proteome Discoverer software as described above.

A total of 14,386 peptides from 26,127 detected peptides were considered quantifiable after removal of low confidence hits and contaminants and were used for further analyses (Supplementary Data 1). Functional and taxonomic annotations of the peptides were derived by mapping back to the corresponding proteins. Peptide abundances for peptides mapping to a taxonomic or functional category were summed to identify trends of the category with solids loadings. In-house scripts were used to ascertain the taxonomic source of a peptide if it mapped to >1 protein with different taxonomic origins based on GhostKOALA assignment. Peptides were qualified as microbial (Bacteria or Archaea), plant (Switchgrass or other plant sequences), or mixed origin (indistinguishable). Peptides from cellulolytic organisms were determined if they mapped to proteins from CAZymes as determined by dbCAN2. Peptides from dockerin were determined from InterProScan. Peptides from methanogenic organisms were determined if they belonged to *Euryarchaeota* proteins. Pairwise comparisons for each solids loadings against the 30 g/L condition were conducted using Welch's $t$-test and a cutoff of $p$-value ≤ 0.05 was used. Python and R libraries were used to generate figures.

**Reporting summary**. Further information on research design is available in the Nature Research Reporting Summary linked to this article.

## Data availability

The data on solubilization and microbiome performance generated in this study is available in the supplementary information files. All proteomics raw mass spectra used for protein quantification in this study are available at the ProteomeXchange Consortium via the MassIVE repository (MassIVE accession: MSV000088319 [https://massive.ucsd.edu/ProteoSAFe/dataset.jsp?task=05b15f47bc0145759b12f5da310d3a6a]; ProteomeXchange accession: PXD029582). All proteome abundance data along with the mapped annotations are available as Supplementary Data files. Source data are provided with this paper.

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

## Acknowledgements

Oak Ridge National Laboratory: Jason Witham (bioinformatics), Dawn M. Klingeman (sample preparation), Steven D. Brown (coordination) and James Elkins (microbial community insights). The Pennsylvania State University: Tom Richard (microbial enrichment insights). Dartmouth College: Xiongjun Shao (assistance in experimental and method design), Sean Murphy, Jules Wheaton, Lion Herfort, Liang Tian and Anela Arifi (assistance with sample analysis and microbiome operation). Funding was provided by the BioEnergy Science Center and the Center for Bioenergy Innovation, both at the U.S. Department of Energy (DOE) Research Center supported by the Office of Biological and Environmental Research in the DOE Office of Science.

## Author contributions

E.K.H., L.R.L, P.C. R.J.G., and R.L.H. designed the study. X.L. and E.K.H. performed and analyzed the fermentation experiments. P.C., S.P., and R.J.G. conducted the metaproteomics measurements. J.C.E. co-assembled the metagenomes and generated the metaproteome database. P.C., R.J.G, and R.L.H performed all the metaproteomics data analyses. E.K.H. and Y.J.B. helped in the interpretation of CAZyme results, Y.J.B. co-wrote the paper. P.C., E.K.H., R.J.G., X.L., R.L.H. and L.R.L. wrote the paper. All authors edited and reviewed the paper.

## Competing interests

L.R.L. is a shareholder in a startup company focusing on cellulosic biofuel production. All other authors declare no competing interests.
