## [Peer Review File · Nature Communications]

Metaproteomics reveals enzymatic strategies deployed by anaerobic microbiomes to maintain lignocellulose deconstruction at high solidsReviewers' Comments:

Reviewer #1:

Remarks to the Author:

The present manuscript provides insights into the mechanisms of lignocellulose conversion by microbial communities at high solids loadings. It represents a significant scientific contribution by reporting the metaproteomic characterization of a microbiome as a function of substrate loading across different fractions, depicting CAZyme content and methanogenesis features.

Overall, the work seems to have been well performed using suitable experimental techniques. The present manuscript brings an interesting point that should interest others working in the same field. While the manuscript outlines exciting observations, there are points to be clarified which are essential for the merit of this publication:

It is vague how the present work advances from a previous study from Liang et al. (<https://doi.org/10.1186/s13068-018-1238-1>); this aspect jeopardizes the novelty of the present work.

The switchgrass was added without sterilization, so new microorganisms were inoculated every semi-continuously fed cycle. According to the previous work from Liang et al., it is observed variation of the phylogenetic composition of the microbial community, which changes from different reactors and in response to solids loading. These characteristics could impact the reproducibility of the system and impart biotechnological applicability.

Along with AA6, superoxide dismutases (SOD2) and superoxide reductases (SOR) increased in abundance with increasing solids in all three fractions. It is a pretty intriguing result because it is possible to suppose that hydrogens peroxide are being depleted by SOD and SOR, which would impair the production of Fenton products.

It would be necessary to evaluate hydroxyl radicals or hydrazine production levels to support the statement that Fenton chemistry could be the primary mechanism to explain the high efficiency at high solid loadings.

The data may provide insights into cellulosomes' contribution to microbial degradation at high solid loading.

Figure S4 and S13 contents are not visible.

The phylum-resolved abundance trends of proteins annotated as AA6 should be included in Figure S10.

Finally, it is challenging to picture how this thermophilic, lignocellulose-fermenting, methanogenic anaerobic microbiome could be used to produce various value-added biochemicals and biofuels from plant biomass.

Reviewer #2:

Remarks to the Author:

This is a very interesting paper using the under-utilised metaproteome to investigate metabolic pathways and other microbial processes that are involved in lignocellulose breakdown. The impact of higher solids to overall performance is not necessarily widely understood, and perhaps could be emphasised more in the introduction. The work is closely linked to a metagenomics study, and it

would be have been interesting to see how they compare in terms of predicted function(gene level) and actual function(protein level). The paper was an enjoyable read, but there are a few aspects that need improvement, there are a few missing bits of information (edited on manuscript) and typos that can easily be corrected. My only major issue is supportive information- as proteomics is conducted to allow us to make hypothesis about cellular processes, rather than measuring the actual process itself, it is common to perform corroborating, or at least complementary experiments that can support your findings. This doesn't appear to have been done, e.g. enzyme assays, blots etc. Although the workflow is sound, the need to support with these types of experiments cannot be over-stated, particularly for a journal specific to proteomics, or of the standard of Nature Communications.

Metaproteomics reveals enzymatic strategies deployed by anaerobic microbiomes to maintain lignocellulose deconstruction at high solids

REVIEWER COMMENTS

Reviewer #1 (Remarks to the Author):

The present manuscript provides insights into the mechanisms of lignocellulose conversion by microbial communities at high solids loadings. It represents a significant scientific contribution by reporting the metaproteomic characterization of a microbiome as a function of substrate loading across different fractions, depicting CAZyme content and methanogenesis features.

Overall, the work seems to have been well performed using suitable experimental techniques. The present manuscript brings an interesting point that should interest others working in the same field. **While the manuscript outlines exciting observations, there are points to be clarified which are essential for the merit of this publication:**

Fair point – we have tried to attend to each of these, as detailed below.

It is vague how the present work advances from a previous study from Liang et al. (<https://doi.org/10.1186/s13068-018-1238-1>); this aspect jeopardizes the novelty of the present work.

We appreciate the point raised by the reviewer to improve clarity. There are three main points where this submitted manuscript differs from Liang et al., 2018: (1) the submitted manuscript focuses on metaproteomic characterization; Liang et al. includes no metaproteomics. (2) The submitted manuscript explores increasing solids loadings at a fixed residence time while Liang et al. explored different residence times for one fixed solids loading. (3) Additionally, the submitted manuscript looks at three different fractions of the broth: supernatant (cell free), suspended cells, and pelleted fraction. It provides unprecedented insights in how the microbiome adjusts to accomplish sustained solubilization at high solids across important cellular locations. Neither of these insights were offered in the Liang et al. paper.

We have emphasized this difference and complementarity by adding the following text to the introduction section of the manuscript (page 3, lines 91-96):

“The data and results described in this manuscript are an extension of the previous work described in Liang et al.⁴⁴, where different residence times were examined (20 to 3.3 days) at one fixed solids loading of 30 g/L. The resulting metagenomes from that work were used as a basis for the new metaproteomics analysis described in this paper where the residence time was fixed at 10 days, with increasing solids loading from 30 to 75, 120, and finally 150 g/L of the same feedstock.”

The switchgrass was added without sterilization, so new microorganisms were inoculated every semi-continuously fed cycle. According to the previous work from Liang et al., it is observed variation of the phylogenetic composition of the microbial community, which changes from different reactors and in response to solids loading.

Metaproteomics reveals enzymatic strategies deployed by anaerobic microbiomes to maintain lignocellulose deconstruction at high solids

These characteristics could impact the reproducibility of the system and impart biotechnological applicability.

We have looked at the possible contribution of microorganisms from the feedstock material in Liang et al., 2018 (Figure 4A & B, right column Substrate (S) profile). While these data are derived from abundance of 16S rDNA measurements and do not relay absolute quantitative information, the contribution from the feedstock material was minimal. The same exact feedstock, from a large, well preserved, and maintained stockpile, was used in the current manuscript.

Please note that different solids loadings were not described in Liang et al. While there was indeed phylogenetic variability between reactors in Liang et al., 2018 (Figure 4), that did not translate into functional variability between the reactors as was shown by the low variability in solubilization and methanogenesis results in figures 1a, c, 2, and 3 (Liang et al., 2018).

To address this point, we have added the following sentence on page 13, lines 417-418:

“Potential microbial contributions via the addition of unsterilized feedstock to the microbiome were investigated by Liang et al⁴⁴, and found to be small.”

Along with AA6, superoxide dismutases (SOD2) and superoxide reductases (SOR) increased in abundance with increasing solids in all three fractions. It is a pretty intriguing result because it is possible to suppose that hydrogens peroxide are being depleted by SOD and SOR, which would impair the production of Fenton products.

We are equally intrigued and excited about this finding, as it is derived from multiple lines of evidence. We are pleased that this point and its significance are clear to the reviewer.

It would be necessary to evaluate hydroxyl radicals or hydrazine production levels to support the statement that Fenton chemistry could be the primary mechanism to explain the high efficiency at high solid loadings.

While we believe the observed efficiency at high solid loadings is a result of multiple strategies and that Fenton chemistry is one of the more striking and intriguing routes employed by the microbiome to assist in the solubilization at high solid loadings along with other enzymatic reactions, we want to be careful not to state or even imply that this is at this point known to be the “primary mechanism.” We have modified the text (page 12, lines 382-385) to capture this point:

“Although further investigation is needed in the mechanism proposed here, the observations suggest that AA6 plays an important synergistic role with other CAZyme categories for high solids deconstruction under anaerobic thermophilic conditions.”

Because of the formidable challenge of directly and accurately measuring hydroxyl radicals in a microbiome embedded in a complex matrix, we felt that a compelling route would be to monitor a key suite of enzymes that indicate reactive radical presence/activity. To this end, we examined the expression of several enzymes (Bfr, SOD2, SOR) which would be produced in response to the stress of reactive species. We found that these complementary enzymes (represented by

Metaproteomics reveals enzymatic strategies deployed by anaerobic microbiomes to maintain lignocellulose deconstruction at high solids

multiple proteins from multiple taxa performing a particular enzymatic function) follow the same abundance trend, suggesting the presence of reactive species in the system.

The data may provide insights into cellulosomes' contribution to microbial degradation at high solid loading.

We agree that examining the contribution from cellulosomes at high solids loading is interesting, accordingly, we examined CAZymes with CBMs of cellulosomal domains, as highlighted in following sentences (page 6, lines 177-182):

“About twenty percent (110 of 551) of the CAZymes harbored a carbohydrate binding domain (CBM) or a cellulosomal domain (cohesin or dockerin) (Fig. S6, Supplementary Table S8). The proportion of these affinity-conferring CAZymes declined in the SNT fraction (from ~60% to ~20%) with increasing solids, while minimal changes were observed in the PC (~30%) and SB (~50%) fractions (Fig. S6, S7), suggesting that free enzymes are somewhat excluded from binding substrate at lower solids loadings and remain in the SNT until surfaces become available.”

Figure S4 and S13 contents are not visible.

Agreed. The figures have been revised to improve visibility.

The phylum-resolved abundance trends of proteins annotated as AA6 should be included in Figure S10.

We have changed figure S10 as recommended.

Finally, it is challenging to picture how this thermophilic, lignocellulose-fermenting, methanogenic anaerobic microbiome could be used to produce various value-added biochemicals and biofuels from plant biomass.

We agree with the reviewer that it is challenging to have a mixed microbial community produce a particular value-added chemical or biofuel, with methane as an obvious exception. While the production of value-added chemicals and biofuels from plant biomass via mixed microbial communities has been the topic of several research efforts, our direct interest is particularly to gain insights into Nature's ability to process lignocellulose (in the form of this microbiome). It is not our aim to produce biofuels with a microbiome, but to learn from them, as potential sources of “novel” biocatalysts, and as a benchmark what Nature is able to do under the chosen conditions.

We have clarified this point in the text (page 3, lines 84-86):

“In this study, we employ LC-MS/MS-based metaproteomic measurements in order to gain insight into mechanisms of lignocellulose deconstruction at solids loadings representative of those anticipated in an industrial process.”

Metaproteomics reveals enzymatic strategies deployed by anaerobic microbiomes to maintain lignocellulose deconstruction at high solids

Reviewer #2 (Remarks to the Author):

This is a very interesting paper using the under-utilised metaproteome to investigate metabolic pathways and other microbial processes that are involved in lignocellulose breakdown. **The impact of higher solids to overall performance is not necessarily widely understood, and perhaps could be emphasised more in the introduction.**

Thank you for pointing out the need to emphasize the overall importance of high solids for the wider audience. We have revised the manuscript by adding the following text and a new reference to underscore the need and performance improvements provided by higher solids compared to lower solids (page 2, lines 41-47):

“Because substantial titers of liquid fuel products are required to avoid high costs for product recovery and fermentation, biological processes for conversion of lignocellulose need to operate at high solids loadings – typically on the order of 15 wt.% or more⁵⁻⁷. Around two-thirds of the mass content of lignocellulose is carbohydrate. An efficient sugar-to-liquid-biofuel microbial metabolism can achieve an end-product at 50% yield. Not considering titer restrictions and solids handling issues, 150 g/L solids loading would result in a maximum biofuel titer for ethanol of ~50 g/L.”

The work is closely linked to a metagenomics study, and it would be have been interesting to see how they compare in terms of predicted function(gene level) and actual function(protein level).

Yes, we agree that under different circumstances it would be interesting since the metagenomics analysis in Liang et al., 2018 provided the basis for the metaproteomics results discussed in the current manuscript. However, as the current manuscript reports on different conditions than those reported in Liang et al, and focuses solely on metaproteomics, no additional metagenomics data was generated for these conditions and thus comparing predicted vs. actual function at different solids loadings was out of scope here.

The paper was an enjoyable read, but there are a few aspects that need improvement, there are a few missing bits of information (edited on manuscript) and typos that can easily be corrected.

We appreciate the edits from the reviewer. We have made the requested edits in the manuscript (marked in the version with track changes) to make the manuscript clearer.

My only major issue is supportive information- as proteomics is conducted to allow us to make hypothesis about cellular processes, rather than measuring the actual process itself, it is common to perform corroborating, or at least complementary experiments that can support your findings. This doesn't appear to have been done, e.g. enzyme assays, blots etc. Although the workflow is sound, the need to support with these types of experiments cannot be over-stated, particularly for a journal specific to proteomics, or of the standard of Nature Communications.

Metaproteomics reveals enzymatic strategies deployed by anaerobic microbiomes to maintain lignocellulose deconstruction at high solids

While corroborating experiments would certainly be of interest, we believe that our paper as revised robustly meets the standard of Nature Communications for novelty and comprehensiveness. Because the resulting datasets in this paper are comprised of a multitude of proteins and pathways that are connected in somewhat complex networks, it is difficult to envision how a few specific assays would reflect the myriad of processes at work in this system. Given the complexity of the feedstock, microbial community, and reaction networks in the system we studied, confirmation of proposed mechanisms would certainly be the next distinct research element that is now attackable based on these results but is beyond the scope of this manuscript. In conceiving our study, it was a priority to ask and answer a specific question about overall microbiome community function. In particular, we carefully designed and executed a systems-biology research campaign to ask – for the first time at this level of detail - “How does an anaerobic microbiome customize its suite of molecular machinery in response to increasing solids loading?” This question is answered in a way that is clear, comprehensive, and unexpected and unforeseeable in several important respects. As it is, it is a challenge for us to present – and we suspect for a reader to assimilate – our extensive data and interpretations within the formatting constraints for a research article in Nature Communications. Adding corroborating experiments at this stage would exacerbate this.

Additional Edits:

The following additional edits were made and are present in the manuscript version with tracked changes:

1. Subheadings in the “Result” section were modified to be ≤60 characters per the formatting requirements.
2. Main figures were updated to meet the font size and resolution requirements.
3. The “Data Availability” section was updated as there were accessibility issues during initial submission.
4. In Line 402, reference for Kubis et al. was updated from “*Kubis et al., under review*” to ref 88 as it has been published since the earlier submission.

Reviewers' Comments:

Reviewer #1:

Remarks to the Author:

The authors have properly addressed all my points raised. It is a well-performed and relevant study.

Reviewer #2:

Remarks to the Author:

Thank you for the revisions, I am happy to recommend acceptance.